# Track3R: Joint Point Map and Trajectory Prior for Spatiotemporal 3D Understanding

**Seong Hyeon Park**
KAIST
seonghyp@kaist.ac.kr

**Jinwoo Shin**
KAIST and RLWRLD
jinwoos@kaist.ac.kr

## Abstract

Understanding the 3D world from 2D monocular videos is a crucial ability for AI. Recently, to tackle this underdetermined task, end-to-end 3D geometry priors have been sought after, such as pre-trained point map models at scale. These models enable robust 3D understanding from casually taken videos, providing accurate object shapes disentangled from uncertain camera parameters. However, they still struggle when affected by object deformation and dynamics, failing to establish consistent correspondence over the frames. Furthermore, their architectures are typically limited to pairwise frame processing, which is insufficient for capturing complex motion dynamics over extended sequences. To address these limitations, we introduce Track3R, a novel framework that integrates a new architecture and task to jointly predict point map and motion trajectories across multiple frames from video input. Specifically, our key idea is modeling two disentangled trajectories for each point: one representing object motion and the other camera poses. This design not only can enable understanding of the 3D object dynamics, but also facilitates the learning of more robust priors for 3D shapes in dynamic scenes. In our experiments, Track3R demonstrates significant improvements in a joint point mapping and 3D motion estimation task for dynamic scenes, such as 25.8% improvements in the motion estimation, and 15.7% in the point mapping accuracy.

## 1 Introduction

The recent advance of 3D prior models by the point mapping frameworks has enabled robust and accurate 3D understanding in casually taken video frames. Unlike the 3D reconstruction methods that rely on external matching, depth, and pose estimation priors [1], these models directly learn 3D shape priors via end-to-end designs predicting dense 3D points directly from 2D frames. Notably, DUSt3R [2] has introduced the pair-wise point mapping, which, given a pair of two image frames with unknown camera parameters, maps every pixel in one frame to 3D points in the other frame's view. This design allows robust 3D representation disentangling the 3D shapes and the camera motion, providing a strong prior trained at scale.

However, current methods are often challenged when given videos capturing dynamic scenes, affected by moving and deforming objects. For example, the point map task accounts for only variable camera poses, but cannot establish a consistent motion trajectory of the 3D points over the frames. Furthermore, the typical model architectures suffer from the constrained temporal window size of 2 frames, which hinders modeling complex dynamics spanning over wider windows. Although there have been approaches to mitigate the problem, such as injecting motion estimation priors [3], memory bank architecture [4], they cannot generalize well to various tasks in dynamic scenes and fail to learn holistic priors that can disentangle the object dynamics.

To tackle this problem, we propose a new framework which can jointly predict the point map and their motion trajectory from a multiple number of frames in the end-to-end manner, coined Track3R.

39th Conference on Neural Information Processing Systems (NeurIPS 2025).

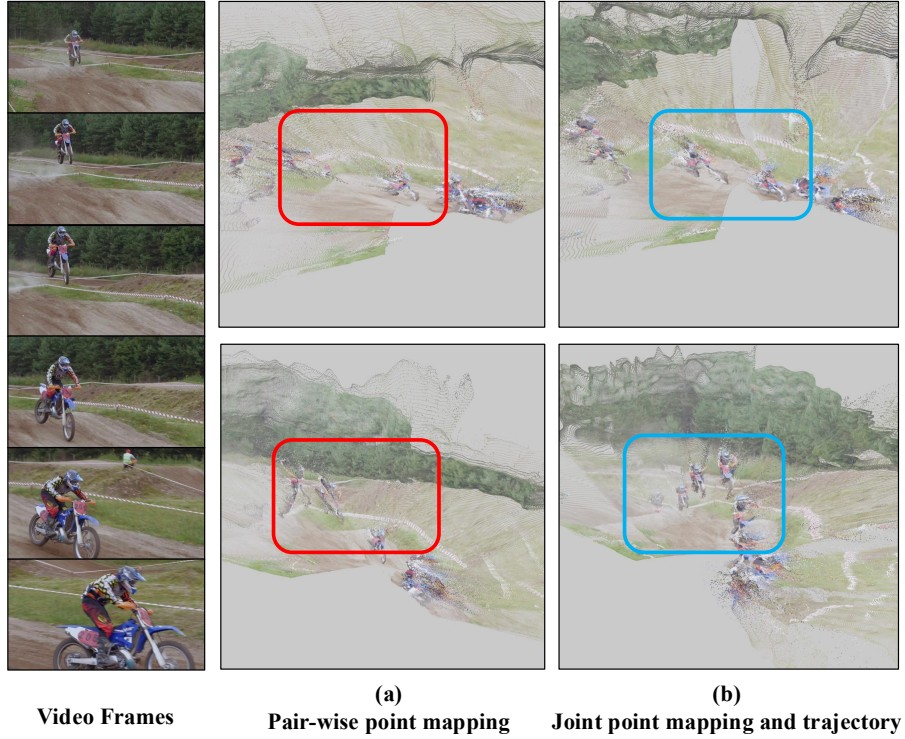

| | **(a)** | **(b)** |
|---|---|---|
| **Video Frames** | **Pair-wise point mapping** | **Joint point mapping and trajectory** |

Figure 1: **The joint point mapping and trajectory prediction.** The outputs of a pair-wise point mapping [3] and our method are compared, processing 6 distant frames of a video [5] capturing dynamic scenes. With the first frame as the reference, the 3D points are predicted for other frames.

Specifically, our key idea is modeling two disentangled trajectories for each point: one representing the 3D motion and the other representing camera views. We enable this task via generalizing a well-adopted point mapping framework, the Siamese transformer architecture with two parallel decoders [6, 7] for the pair-wise point mapping, where the first and the second decoders process distinct frames.

To be specific, assuming $W$ input frames, we introduce a factorized temporal attention over the tokens in the same spatial locations of the frames [8], which enables predicting point maps beyond the 2-frame constraint. For instance, while the vanilla architecture would perform `Permutation`$(W, 2)$ pair-wise iterations, our enhanced architecture would produce an equivalent set of point map sequences in a single forward pass,[1] as depicted in Figure 1. Then, we modify the semantics represented by these sequences: the output by the first decoder represents the 3D motion trajectory in a fixed camera coordinate system, and the other by the second decoder to be 3D points in the variable camera views.

Our model can readily utilize 3D shape priors pre-trained on static scenes at scale, which enables a training focused on fine-tuning the model in dynamic scenes for learning motion priors. For example, we employ 3D trajectory annotations from human motion datasets [9] as the ground truth for training, and synthesize the input frames via a dedicated 3D rendering framework for the human bodies [10]. While the dynamic video data in prior art, such as Kubric [11] and Point Odyssey [12], are with a sparse set of trajectory annotations, human motion datasets provide a denser set of point trajectories, which can facilitate learning stronger priors for joint 3D geometry and motion in dynamic scenes.

Track3R not only can provide more accurate outputs for the tasks involving 3D motion estimation, but also enables the learning of more robust priors for 3D geometry in dynamic scenes. For example, it achieves significantly improved experimental results compared to strong baselines, *e.g.*, relative 25.8% on 3D motion estimation and 15.7% on point mapping tasks.

---

[1]We note that the vanilla architecture and the enhanced architecture are essentially identical if $W = 2$.

## 2 Related Work

### 2.1 3D prior learning

The traditional approaches have focused on learning priors for matching and depth features, which only partially describe the 3D configuration in images and prone to the error accumulation in the independent prior models. Since the introduction of end-to-end architectures, such as the image-based neural fields [13] and the feed-forward Gaussian splatting [14], the direct 3D prior learning from image inputs has gained research interest. However, their main focus is typically on reconstructing photometric colors of the image pixels, assuming the camera parameters should be known a priori, which hinders learning robust 3D representation from casually-taken videos [15].

Recently, the point mapping task [2] has been introduced, which choose to directly regress the 3D point coordinates without requiring known camera configurations. The point map can effectively disentangle the influence of camera motion in the 3D geometry, which has been shown to enable learning robust 3D shape priors trained at scale. Our goal is to further generalizing the point mapping 3D prior learning for dynamic scenes, which we further discuss as follows.

### 2.2 3D tracking models

Recently, the conventional 2D video point tracking tasks [16] have evolved towards understanding motion in 3D space [17]. Since the goal of the 3D tracking is predicting the motion trajectory and detecting the occlusions of a query point in video frames, assuming that the camera poses are known a priori, the state-of-the-art methods in this field often propose unifying the 2D video point tracking models, depth estimation, and camera pose estimation methods, *e.g.*, DELTA [18], TAPIP3D [19], and SpatialTracker [20]. We note that these models focus on motion estimation in videos rather than learning holistic prior for 3D shapes, and further discuss the 3D point mapping methods and their extension to dynamic scenes in the follwing subsection.

### 2.3 3D point mapping models

Since the seminal work in DUSt3R [2], which introduced the Siamese decoding architecture for the pair-wise point mapping, numerous contributions have been made to enhance the generalization performance to various 3D modeling scenarios, namely the application to dynamic scenes, and the extended temporal window size for processing multiple frames.

For example, the approaches for fine-tuning from synthetic data is found effective for enhancing the 3D reconstruction in dynamic scenes [3, 21], although they require an external prior models for motion estimation or metric depth estimation. Later, the extended task definitions have been introduced to directly learn priors for the temporal correspondence, or tracking between a fair of input frames [22, 23]. However, these methods are still constrained to the pair-wise architecture and often rely on heavy test-time optimization for processing multiple frames. Although the technique for expediting the optimization is a concurrent area of research [24], the time cost persists to be high.

In order to achieve the multi-frame modeling without relying on the optimization, new architectures have been proposed, such as the memory bank [4, 25], the multi-view cross attention [26], non-Siamese decoders [27], and employing DINO [28] features as additional inputs [29]. However, the task considered by these models cannot explicitly handle motion in video frames, which hinders generalization to dynamic videos. While our framework also considers an enhanced architecture with the modified attention, we additionally generalize the task for a joint point mapping and trajectory prediction, which has significant effect for understanding dynamic scenes.

It is also worth noting concurrent works such as GFlow [30], POMATO [31], St4RTrack [32], which have been devoted to tackle handling dynamic videos in point mapping architectures. For example, GFlow [30] employs the Gaussian Splatting [33] for better optimization, POMATO [31] proposes a new temporal prediction head for motion estimation, and St4RTrack [32] employs a pair-wise tracking task and architecture.

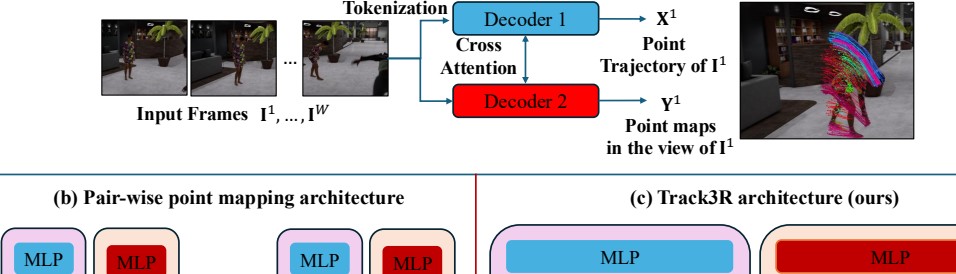

(a) The joint point mapping and trajectory prediction

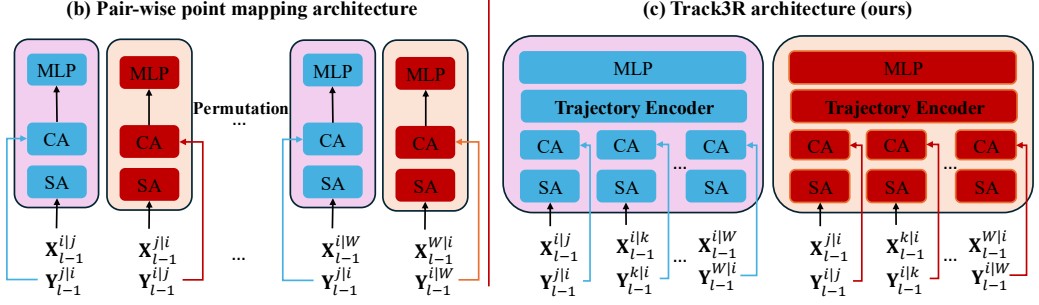

Figure 2: **Illustration of the prediction pipeline in Track3R.** The top figure (a) depeicts the overall pipeline, with the first decoder (Decoder 1) and the second decoder (Decoder 2), shared by both the point mapping frameworks. The bottom-left figure (b) illustrates the design of the decoder blocks in the pair-wise point mapping, using the self-attention (SA) and the cross-attention (CA) mechanisms. The bottom-right figure (c) illustrates our architecture for joint point mapping and trajectory, equipped with the proposed trajectory encoder.

## 3  Method

In this section, we provide the details of our architecture design for point mapping and trajectory prediction given a video sequence. We denote scalars using normal letters, and tensors using bold letters with a superscript denoting frame indices. For example, an input RGB video frame is $\mathbf{I}^i \in \mathbb{R}^{U \times V \times 3}$, where $U \times V$ is the resolution, and a frame tokenization is $\mathbf{F}^i \in \mathbb{R}^{N \times D}$, where $N = \frac{U}{P} \times \frac{V}{P}$ with the patch size $P$ and the embedding dimension $D$. Tensors can be indexed, such as $\mathbf{F}^i(n) \in \mathbb{R}^D$, where $\mathbf{F}^i \equiv [\mathbf{F}^i(1), ..., \mathbf{F}^i(N)]$. Finally, to highlight that a feature or data for frame $i$ is conditioned on the frame $j$, we use the superscript $i|j$, such as the point map $\mathbf{X}^{i|j} \in \mathbb{R}^{U \times V \times 3}$.

### 3.1  Pair-wise point mapping

Given a pair of frames $(\mathbf{I}^i, \mathbf{I}^j)$, the point mapping model aims to predict 2 different 3D points: the first decoder predicting $\mathbf{X}^{i|j}$ which represents the 3D coordinate of $\mathbf{I}^i$, and the second decoder predicting $\mathbf{Y}^{j|i}$ which represents the 3D coordinate of $\mathbf{I}^j$ in the camera view of $\mathbf{I}^i$. Specifically, the vanilla Siamese architecture [2] employs transformer blocks with cross-attention. To describe the operations within a block, we denote the tokens in the first decoder as $\mathbf{X}^{i|j}_l$, and the second decoder as $\mathbf{Y}^{i|j}_l$, where $l \in \{0, ..., L\}$ is the block index, with the initialization $\mathbf{X}^{i|j}_0 := \mathbf{F}^i$ and $\mathbf{Y}^{j|i}_0 := \mathbf{F}^j$.

In each transformer block (Figure 2b), the cross-attention $\mathtt{CA}(\cdot; \cdot)$, placed next to the self-attention $\mathtt{SA}(\cdot)$, conveys information between the two decoders, followed by the $\mathtt{MLP}(\cdot)$ layer, producing the parallel outputs,

$$\tilde{\mathbf{X}}^{i|j}_l := \mathtt{CA}\big(\mathtt{SA}(\mathbf{X}^{i|j}_{l-1}); \mathbf{Y}^{j|i}_{l-1}\big) \quad \text{and} \quad \tilde{\mathbf{Y}}^{j|i}_l := \mathtt{CA}\big(\mathtt{SA}(\mathbf{Y}^{j|i}_{l-1}); \mathbf{X}^{i|j}_{l-1}\big) \tag{1}$$

$$\mathbf{X}^{i|j}_l := \mathtt{MLP}\big(\tilde{\mathbf{X}}^{i|j}_l\big) \quad \text{and} \quad \mathbf{Y}^{j|i}_l := \mathtt{MLP}\big(\tilde{\mathbf{Y}}^{j|i}_l\big), \tag{2}$$

assuming the skip-connections [34, 35] existing in the layers. To produce the final outputs, the DPT head layer [36] is employed, which takes these parallel block-wise tokens as the input,

$$\mathbf{X}^{i|j} := \mathtt{Head}\big(\mathbf{X}^{i|j}_0; \mathbf{X}^{i|j}_1; ...; \mathbf{X}^{i|j}_L\big) \quad \text{and} \quad \mathbf{Y}^{j|i} := \mathtt{Head}\big(\mathbf{Y}^{j|i}_0; \mathbf{Y}^{j|i}_1; ...; \mathbf{Y}^{j|i}_L\big). \tag{3}$$

Although we abuse the same notations SA, CA, MLP, and Head for the two decoders and for all block indices $l \in \{1, ..., L\}$, we note that their weight parameters are all different.

This pair-wise point mapping model is typically executed twice, for the permutation of the input frames, which enables downstream tasks, such as 2-view geometry, estimating camera poses, etc. When processing a larger number of frames, *e.g.*, $W > 2$, the inference is performed over Permutation$(W, 2)$ for all $i \in \{1, ..., W\}$ and $j \in \{1, ..., W\}$.

## 3.2 Generalized point trajectory prediction

As discussed in the previous section, the point mapping on $W$ frames would essentially produce two sets of point map sequences per frame $\mathbf{X}^W \in \mathbb{R}^{(W-1) \times U \times V \times 3}$ and $\mathbf{Y}^W \in \mathbb{R}^{(W-1) \times U \times V \times 3}$. For example, assuming $W = 3$ input frames, $\{\mathbf{I}^1, \mathbf{I}^2, \mathbf{I}^3\}$, $\mathbf{I}^3$ would be associated with the set of point maps from the first and second decoders, respectively:

$$\mathbf{X}^3 := [\mathbf{X}^{3|1}, \mathbf{X}^{3|2}] \quad \text{and} \quad \mathbf{Y}^3 := [\mathbf{Y}^{1|3}, \mathbf{Y}^{2|3}]. \tag{4}$$

Given these sequences, one can quickly notice that the outputs from the first decoder, $\mathbf{X}^{3|1}$ and $\mathbf{X}^{3|2}$, represent the same semantics: the 3D coordinate of $\mathbf{I}^3$ in the camera view of $\mathbf{I}^3$. Our key idea is to resolve this duplication, by considering a modified definition, $\mathbf{X}^{3|1}$ and $\mathbf{X}^{3|2}$ representing the 3D coordinate of $\mathbf{I}^3$ in the frame index of $\mathbf{I}^1$ and $\mathbf{I}^2$, respectively, in the fixed camera view of $\mathbf{I}^3$. That is, we predict the 3D motion trajectory of the frame $\mathbf{I}^3$.

In order to learn this new task, we employ the confidence-aware regression loss from the original point mapping literature [2], yet provide the 3D trajectory annotations as the true targets for $\mathbf{X}$'s, which have been relatively sparse in datasets considered by prior art. Therefore, we further facilitate the learning by utilizing human motion dataset, which provide a denser motion trajectory and human body models [9], where the corresponding video frames can be readily rendered via a dedicated framework [10]. We note that the task definition of $\mathbf{Y}$'s remain unchanged, representing the 3D coordinate of a frame pixels mapped to the views of other frames.

## 3.3 Trajectory encoder

Despite the generalized task, the pair-wise architecture is still constrained to process the complex dynamics spanning $W > 2$. To tackle this problem, we describe our method to jointly process multiple frames (*i.e.*, $W > 2$). Specifically, we enable it with the trajectory encoder module. This module is based on the factorized temporal attention, which collects the tokens in the same spatial index over the frames to encode the inter-frame dynamics back to each token.

Let us consider the frame $\mathbf{I}^W$, paired with others $\{\mathbf{I}^1, ..., \mathbf{I}^{W-1}\}$ and their corresponding tokens within the intermediate cross-attention stage of the decoder blocks in Equation (1),

$$\tilde{\mathbf{X}}_l^{W|\{k<W\}} = \{\tilde{\mathbf{X}}_l^{W|1}, ..., \tilde{\mathbf{X}}_l^{W|W-1}\} \quad \text{and} \quad \tilde{\mathbf{Y}}_l^{W|\{k<W\}} = \{\tilde{\mathbf{Y}}_l^{W|1}, ..., \tilde{\mathbf{Y}}_l^{W|W-1}\}. \tag{5}$$

Intuitively, gathering from a same spatial index, *e.g.*, a stack of tokens $[\tilde{\mathbf{Y}}_l^{W|1}(n), ..., \tilde{\mathbf{Y}}_l^{W|W-1}(n)]$ by indexing each element in Equation (5), can represent the spatio-temporal dynamics of the patch region represented by $\mathbf{F}^W(n)$. Therefore, projecting this feature onto each token can encode the dynamics. Specifically, we apply a factorized attention[2] with causal masks to implement the function, coined trajectory attention TA$(\cdot; \cdot)$,

$$\bar{\mathbf{X}}_l^{W|j} := \text{TA}(\tilde{\mathbf{X}}_l^{W|j}; \tilde{\mathbf{X}}_l^{W|\{k<W\}}) \quad \text{and} \quad \bar{\mathbf{Y}}_l^{W|j} := \text{TA}(\tilde{\mathbf{Y}}_l^{W|j}; \tilde{\mathbf{Y}}_l^{W|\{k<W\}}), \tag{6}$$

where

$$\bar{\mathbf{X}}_l^{W|j}(n) = \text{CA}(\tilde{\mathbf{X}}_l^{W|j}(n); [\tilde{\mathbf{X}}_l^{W|1}(n), ..., \tilde{\mathbf{X}}_l^{W|j}(n)]), \tag{7}$$

$$\bar{\mathbf{Y}}_l^{W|j}(n) = \text{CA}(\tilde{\mathbf{Y}}_l^{W|j}(n); [\tilde{\mathbf{Y}}_l^{W|1}(n), ..., \tilde{\mathbf{Y}}_l^{W|j}(n)]). \tag{8}$$

---

[2]We adjust the relative position embedding to encode a spatial index with the size $D/2$, and a time index with the size $D/2$.

However, naively inserting this layer to each decoder block of a pre-trained Siamese model results in sub-optimal performance after training on dynamic scenes. In fact, prior art finds that retaining strong 3D prior learned from static datasets is crucial for dynamic scenes [3]. Since the trajectory attention deviates the computation graph of a pair-wise model, the model can lose the pre-trained 3D prior. We note that it is also non-trivial to pre-train a multi-frame model from scratch, since the training data for 3D geometry is often a pair of images [2], rather than a video stream data.

To address the problem, we aim to minimize the effect of modification in the initial state of the model. Specifically, inspired by model inflation techniques in video transformers [8, 37], which maintain image prior by attenuating the activation of the temporal attentions, we introduce the layerscale $\mathtt{LS}(\cdot)$ initialized to a very small scalar [38] to the module, referring to the whole layer as the trajectory encoder $\mathtt{TE}(\cdot; \cdot)$,

$$\bar{\mathbf{X}}_l^{W|j} := \mathtt{TE}\big(\tilde{\mathbf{X}}_l^{W|j}; \tilde{\mathbf{X}}_l^{W|\{k<W\}}\big) \tag{9}$$
$$:= \tilde{\mathbf{X}}_l^{W|j} + \mathtt{LS}\big(\mathtt{TA}(\tilde{\mathbf{X}}_l^{W|j}; \tilde{\mathbf{X}}_l^{W|\{k<W\}})\big),$$
$$\bar{\mathbf{Y}}_l^{W|j} := \mathtt{TE}\big(\tilde{\mathbf{Y}}_l^{W|j}; \tilde{\mathbf{Y}}_l^{W|\{k<W\}}\big) \tag{10}$$
$$:= \tilde{\mathbf{Y}}_l^{W|j} + \mathtt{LS}\big(\mathtt{TA}(\tilde{\mathbf{Y}}_l^{W|j}; \tilde{\mathbf{Y}}_l^{W|\{k<W\}})\big).$$

This design ensures that the model is equivalent to the pair-wise architecture in the initial state, retaining the pre-trained 3D prior.

To summarize, we enhance the model architecture for multi-frame, and generalize the task definition (in Section 3.2) of the corresponding outputs. Throughout the training on dynamic scenes, our model initialize from a strong 3D shape prior, and gradually learn to model complex multi-frame dynamics and predict 3D motion estimation, achieving the joint point mapping and trajectory prior.

# 4 Experiment

In this section, we present the experimental details and compare Track3R to state-of-the-art baselines. In Section 4.1, we provide the training details, such as the training dataset, schedules, and hyperparameters. In Section 4.2, we discuss the experimental details, such as the baselines, checkpoints, and the inference configurations. Next, we provide the results on the joint point mapping and trajectory prediction in Section 4.3, which aims to evaluate the quality of the motion and shape prior learned by Track3R in dynamic scenes. Then, we further study the downstream task, *e.g.*, the feed-forward camera pose estimation in Section 4.4, which aims to compare the ability to disentangle the camera motion in dynamic scenes. Finally, we present the ablation study of the proposed method in Section 4.5.

## 4.1 Training details

We initialize the Track3R with the pre-trained weight published by MonST3R [3], a pair-wise point mapping architecture trained on dynamic scenes covered by synthetic datasets [12, 39, 40], starting from the pre-trained DUSt3R [2], which is trained on 8M images that capture the real-world static scenes at scale, such as ScanNet [41] and StaticThings3D [42].

For the fine-tuning, we employ the 3D trajectories and video frames, the combination of SMPL-X human motion trajectory [9] and the associated video frames rendered with BEDLAM [10], along with the publicly available Waymo [43] dataset. Specifically, we supervise the output $X$ from the first decoder with the 3D trajectory annotation, and supervise the output $Y$ from the second decoder with the ground truth depth maps, registered to the world coordinate system based on the annotated camera poses. We employ the confidence-aware and scale-invariant regression loss, following the baselines [2, 3].

We train Track3R for 25 epochs using the AdamW optimizer [44] with 25k clips of length $W = 6$ per epoch, the mini-batch size 16, and the learning rate $1 \times 10^{-4}$. The training takes approximately 36 hours on the system equipped with 8 nVidia A100 GPUs.

Table 1: **Joint 3D motion and shape performance.** The quality of 3D motion estimation ($EPE_{3D}$) and the point mapping (Abs-rel) are compared among Track3R (ours) and the baselines. The baselines that perform additional test-time optimizations are grouped in the top block, and the baselines for feed-forward prediction are grouped in the bottom block. The best scores are highlighted with bold numbers, and the best scores within a model category are highlighted with underlines.

| Method | FPS | iPhone $EPE_{3D}$ | iPhone Abs-rel | Sintel $EPE_{3D}$ | Sintel Abs-rel | Point Odyssey $EPE_{3D}$ | Point Odyssey Abs-rel |
|---|---|---|---|---|---|---|---|
| DUSt3R (2024) | 0.62 | 0.973 | 1.212 | 0.565 | 0.422 | 0.933 | 0.184 |
| MonST3R (2025) | 0.41 | 0.619 | 0.310 | 0.401 | 0.335 | 0.481 | 0.090 |
| MegaSaM (2025) | 0.97 | 0.598 | **0.211** | 0.372 | **0.231** | 0.455 | 0.091 |
| Align3R (2025) | 0.17 | 0.581 | 0.290 | 0.487 | 0.263 | 0.670 | **0.075** |
| Fast3R(2025) | 91.81 | 0.692 | 0.419 | 0.667 | 0.517 | 0.771 | 0.214 |
| Spann3R(2025) | 20.77 | 0.658 | 0.431 | 0.523 | 0.622 | 0.591 | 0.231 |
| CUT3R(2025) | 27.43 | 0.701 | 0.408 | 0.681 | 0.428 | 0.609 | 0.177 |
| **Track3R** (ours) | 23.81 | **0.431** | 0.344 | **0.312** | 0.374 | **0.399** | 0.165 |

Table 2: **Feed-forward camera pose estimation.** The quality of estimating camera translation (ATE and RPE-t) and rotation (RPE-r) are comapred among the feed-forward prediction baselines. The best scores are highlighted with bold numbers, and the second-best scores are highlighted with underlines.

| Method | iPhone ATE | RPE-t | RPE-r | Sintel ATE | RPE-t | RPE-r | TUM-dynamic ATE | RPE-t | RPE-r |
|---|---|---|---|---|---|---|---|---|---|
| Fast3R (2025) | 0.413 | 0.294 | 1.561 | 0.377 | 0.150 | 3.233 | 0.129 | 0.111 | 2.794 |
| Spann3R (2025) | 0.552 | 0.310 | 2.301 | 0.329 | 0.110 | 4.471 | 0.056 | 0.021 | 0.591 |
| CUT3R (2025) | 0.291 | 0.184 | 0.848 | 0.213 | **0.064** | **0.596** | **0.046** | **0.015** | **0.473** |
| **Track3R** (ours) | **0.233** | **0.150** | **0.694** | **0.201** | 0.066 | 0.621 | 0.051 | 0.020 | 0.517 |

## 4.2 Experimental details

We consider 7 different baseline point mapping models, DUSt3R [2], MonST3R [3], MegaSaM [24], Align3R [21], Fast3R [27], Spann3R [25], and CUT3R [4]. We experiment with the checkpoint provided in the official open-source repository hosted by their authors, following the default image processing in each model, *e.g.*, , the input dimensions are, the longer side length of 512 in DUSt3R [2], MonST3R [3], Align3R [21], and Fast3R [27], the longer side length of 672, in MegaSaM [24], and, the square $256 \times 256$ in Spann3R [25].

Unless otherwise specified, we always choose the temporal window size of the inference $W = 6$ for evaluating our method and the baselines. We note that the pair-wise processing baselines are iteratively executed to match the required window size.

To evaluate the feed-forward camera pose estimation in Section 4.4, we employ the weighted Procrustes solver to derive the relative rotation and translation between the frames, and the weighted least squares solver to estimate the camera intrinsic parameters, similar to the experimental configuration in CUT3R [4].

## 4.3 Joint point mapping and 3D motion estimation.

In this section, we evaluate the joint point mapping and 3D motion estimation quality. We employ 3 different test datasets covering dynamic scenes: iPhone dataset [45], Point Odyssey [12], and Sintel [46]. iPhone dataset covers the real-world scene, which originally provides monocular video frames, depth, and camera poses collected with the Lidar and IMU sensors [45]. To obtain the 3D motion trajectory annotation, we utilize the track optimization part of the sophisticated stereo video optimization framework [22]. Point Odyssey and Sintel are synthetic datasets rendered with the 3D engine, which provides the ground truth depth and 3D motion trajectories. We note that the validation sets are utilized for these datasets.

Table 3: **Ablation study.** The effect of trajectory encoder, the joint training objective, and the training with human motion dataset are studied in terms of the quality of 3D motion estimation ($EPE_{3D}$) and the point mapping (Abs-rel).

| Modules | iPhone | | Sintel | | Point Odyssey | |
|---|---|---|---|---|---|---|
| | $EPE_{3D}$ | Abs-rel | $EPE_{3D}$ | Abs-rel | $EPE_{3D}$ | Abs-rel |
| Base Model | 0.803 | 0.879 | 0.695 | 0.737 | 0.980 | 0.281 |
| + Trajectory Encoder | 0.682 | 0.593 | 0.576 | 0.531 | 0.704 | 0.239 |
| + Joint Objective | 0.445 | 0.357 | 0.429 | 0.396 | 0.463 | 0.181 |
| + Human Motion | **0.431** | **0.344** | **0.312** | **0.374** | **0.399** | **0.165** |

To provide a holistic view over both 3D motion and shape qualities, we borrow metrics from literature for each task: the 3D end-point-error ($EPE_{3D}$; a regression quality of 3D motion trajectory) [47] and the absolute relative error (Abs-rel; the accuracy of point mapping to be within a 1.25-factor of the ground truth) [4]. We compare Track3R and the baselines: DUSt3R [2], MonST3R [3], MegaSaM [24], Align3R [21], Spann3R [25], Fast3R [27], CUT3R [4] in Table 1.

To begin with, Track3R demonstrates the strongest results in terms of the quality of 3D motion estimation ($ECE_{3D}$), *e.g.*, relative 25.8% improvement compared to Align3R [21] in iPhone dataset ($0.581 \rightarrow 0.431$). We observe the baselines employing test-time optimizations (upper block in Table 1) can demonstrate overall improved motion estimation performance than the feed-forward prediction methods (lower block in Table 1), except Track3R (ours).

Although the test-time optimization can reinforce overall consistency in point mapping along the frames, and inject external motion prior (*e.g.*, MonST3R [3]), it significantly deteriorates inference speed (FPS). Since our Track3R can directly learn the 3D motion prior during training, instead of the test-time, it can achieve a better motion estimation performance and provides a reasonable inference speed as well, *e.g.*, 23.81 (FPS).

In terms of the point mapping accuracy (Abs-rel), Track3R consistently demonstrates the strongest results among the feed-forward prediction methods, *e.g.*, relative 15.7% improvement compared to CUT3R [4] in iPhone dataset ($0.408 \rightarrow 0.344$). Although there is a gap between the test-time optimizations and ours, the significant improvement compared to the strongest feed-forward baseline (CUT3R [4]) supports the effect of the joint prior learning by our method.

### 4.4 Downstream tasks

In this section, as the downstream application, we evaluate the feed-forward camera pose estimation in dynamic scenes, and provide visualization of the point maps in the world coordinate system.

First, in the feed-forward camera pose estimation, we consider the benchmark with respect to Sintel [46] and TUM-dynamics [48], following the configuration in CUT3R [4] for dynamic scenes, and also the customized iPhone dataset [45] evaluated with the sensory camera poses. In Table 2, we compare Track3R (ours) and the feed-forward prediction baselines: Fast3R [27], Spann3R [25], and CUT3R [4]. Our method achieves the strongest results in iPhone dataset, and a performance comparable to CUT3R [4] on in Sintel and TUM-dynamics datasets. For example, Track3R can demonstrate significant improvements in iPhone dataset, which supports that the 3D prior learn by our method can provide a robust prior that better generalize to the real-world dynamic scenes. We observe that CUT3R [4] particularly performs well on synthetic datasets, which is possible by the pose prediction head trained with the ground truth poses in synthetic datasets. In the extension of our goal to learn joint 3D priors, employing such an idea into Track3R can be interesting future work.

Next, for a qualitative demonstration, we provide the visualization of the 3D points in Figure 3, executed on DAVIS video frames [5]. We find our method tends to demonstrate more consistent point maps over the frames, *e.g.*, the background objects and the scene are consistently depicted, comparing the pair-wise point mapping baseline [3] (with red boxes) and our method performing joint point mapping and trajectory prediction (with blue boxes). This supports the significance of our method, which facilitates learning useful representation for predicting accurate point maps as well, even if affected by complex dynamic scenes.

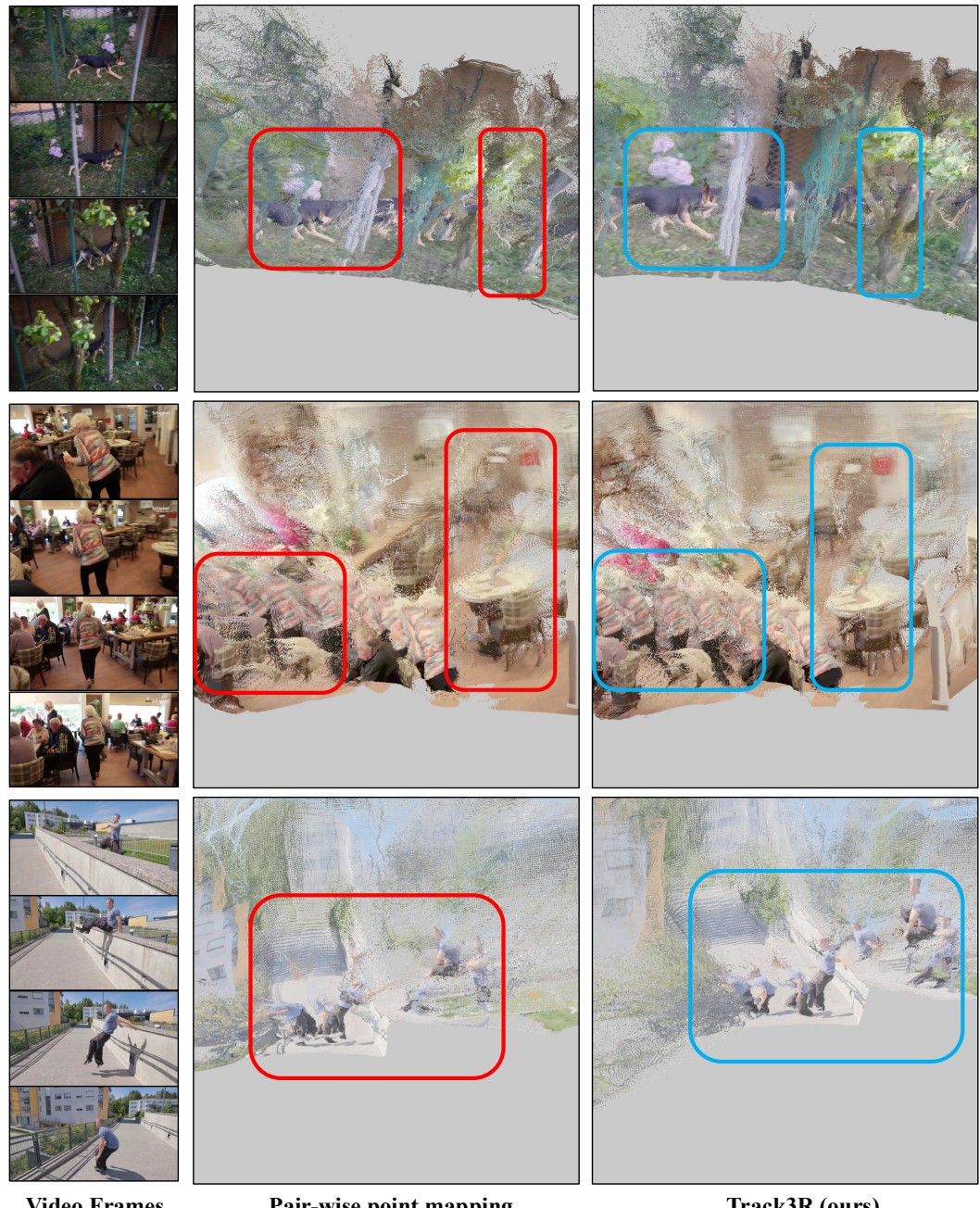

| Video Frames | Pair-wise point mapping | Track3R (ours) |

Figure 3: **Qualitative comparison of 3D points in dynamic scenes.** The 3D points predicted the pair-wise point mapping baseline [3] and Track3R (ours) are compared, using DAVIS video samples [5]. Dynamic regions are highlited with the red and blue boxes.

## 4.5 Ablation study

In this section, we conduct ablation study of the proposed techniques, namely the trajectory encoder, the joint point mapping and trajectory prediction, and the training with human motion data, comparing $EPE_{3D}$ (3D motion estimation) and Abs-rel (point mapping) in Table 3. Overall, we find employing each module is significant to the performance of Track3R, where the flexibility to process multi-frame dynamics is enabled by the trajectory encoder, a robust prior learning is achieved via the joint point mapping and motion estimation objective, and additional performance boosts are observed when trained on human motion.

Although the pair-wise architecture [2, 3] can produce pointmaps for more than 2 frames by executing multiple pair-wise inferences, its design inevitably enforces the assumption that the distributions of consecutive pointmaps are independent. For example, given $\{\mathbf{I}^i, \mathbf{I}^j, \mathbf{I}^k\}$, a pair-wise model assumes that a joint density $\mathtt{Pr}(\mathbf{Y}^{i|j}, \mathbf{Y}^{i|k}, \mathbf{Y}^{j|k})$ is proportional to $\mathtt{Pr}(\mathbf{Y}^{i|j}) \cdot \mathtt{Pr}(\mathbf{Y}^{i|k}) \cdot \mathtt{Pr}(\mathbf{Y}^{j|k})$. However, in practice, including the scenarios represented by our evaluation, there exists an extreme case where $\mathbf{I}^i$ and $\mathbf{I}^k$ are completely non-overlapping, so that the pair-wise model assigns an erroneous estimate of $\mathtt{Pr}(\mathbf{Y}^{i|k})$, which can induce significant failure modes of estimating the joint density. Since Track3R can relax this constraint for multiple frames, it can learn the joint point mapping and trajectory prior that is more close to the true nature of the dynamic scenes.

## 5  Discussion

In this section, we discuss the limitation of Track3R and the future research directions in Section 5.1. We also discuss the potential negative societal impact in Section 5.2.

### 5.1  Limitation

Despite the promising results demonstrated by Track3R, the scarcity of dynamic scenes for training can hinder the generalization performance. To mitigate the problem, we employ the human motion dataset for training. However, the synthesized video inputs can make a distribution shift in the visual texture learned in a pre-trained model. Therefore, designing new training datasets, self-supervised learning with unlabeled data, or an objective functions robust to the distribution shift can be interesting future directions. It is also worth noting that we focus on the realistic scenarios where the observation is captured by a monocular video camera, rather than multiple synchronized cameras capturing one scene. Although it would be straightforward to apply Track3R for the synchronized cameras, we believe that there is a room to exploit useful properties, such as epipolar geometry [49] of the synchronized cameras, which is another interesting future direction.

### 5.2  Potential negative societal impact

While the joint point mapping and motion estimation by Track3R can be beneficial for various video understanding applications, such as novel-view synthesis, depth estimation, and action recognition, the emergence of unexpected behavior within Track3R can lead to misrepresentations of the real video data. For those applications that require extremely accurate models for safety-related judgments, such as depth estimation for autonomous driving, the unexpected behaviors must be carefully managed. To ensure the reliability of systems using point tracking predictions, we recommend to conduct thorough investigations and implement robust mitigation strategies to minimize potential risks, thereby increasing the overall safety and effectiveness of these applications.

## 6  Conclusion

In this paper, we propose Track3R, a joint point mapping and motion estimation framework for learning holistic 3D priors dynamic scenes. We tackle the limitations in existing point mapping baselines, sub-optimal under complex dynamic scenes. For example, we propose to encode the dynamics of the 3D points over multiple frames beyond the pairs, and generalize the task definition to predict the 3D motion trajectories, as well as the static point maps. Our method significantly improves the expressiveness of the model architecture for dynamic scenes, and enables predicting the disentangled representation of the 3D motion and shapes. In the experiments, we find our method can outperform the baselines, for both the 3D motion estimation task and the point mapping task. Overall, our work highlights the effectiveness of jointly solving 3D geometry and motion tasks, and we believe our work could inspire researchers to further leverage it in the future.

**Acknowledgements**. This work was supported by Institute for Information & communications Technology Promotion(IITP) grant funded by the Korea government(MSIT) (No.RS-2019-II190075 Artificial Intelligence Graduate School Program(KAIST); No. RS2024-00509279, Global AI Frontier Lab), the National Supercomputing Center with supercomputing resources including technical support KSC-2025-CRE-0435, and RLWRLD, Inc.

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
