# OpenReview forum: "Track3R: Joint Point Map and Trajectory Prior for Spatiotemporal 3D Understanding"
_NeurIPS.cc/2025/Conference — NeurIPS 2025 poster_

### Official Review · Reviewer_p1Bw · 2025-06-22

**Clarity:** 2
**Significance:** 3
**Originality:** 3
**Rating:** 4
**Confidence:** 4

**Summary:**

Track3R introduces a framework for joint point mapping and motion trajectory prediction from monocular videos, modeling separate object motion and camera pose trajectories across multiple frames. It reports 25.8% improvement in motion estimation and 15.7% in point mapping accuracy over feed-forward baselines in dynamic scenes.

**Questions:**

Why does Track3R underperform test-time optimization methods in point mapping? What specific limitations (e.g., model capacity, training data) cause this? A clear plan to narrow this gap could improve my rating.
Synthetic human motion data may not reflect diverse real-world scenes. Can you provide evidence of performance on varied, non-synthetic data? This could address quality concerns.

**Ethical Concerns:**

["NO or VERY MINOR ethics concerns only"]

**Final Justification:**

The rebuttal from the authors for my review and also for other reviews from other reviewers address my concern about generalizability of the method to non synthetic data. I increased my score accordingly and this paper now fall in the region for borderline accept.

**Limitations:**

yes

**Quality:**

2

**Strengths And Weaknesses:**

**Strengths**: The joint task of predicting point maps and motion trajectories is a meaningful advance, extending pairwise methods to multi-frame processing (Section 3). The trajectory encoder’s factorized attention is a clever adaptation for capturing dynamics (Section 3.3). Experiments on iPhone, Sintel, and Point Odyssey datasets show Track3R outperforms feed-forward baselines in 3D motion (EPE3D) and point mapping (Abs-rel) tasks (Table 1, Section 4.2). The ablation study (Table 3) clearly validates each component’s contribution. The approach has strong potential for 3D vision applications like autonomous driving or AR, where dynamic scene understanding is key (Section 5.2). The paper is well-organized, with helpful figures (Figure 2) and a detailed training setup (Section 4.1). Visualizations (Figure 3) effectively show qualitative gains.


**Weaknesses**: Track3R trails test-time optimization methods (e.g., MegaSaM, MonST3R) in point mapping accuracy (Table 1), limiting its claim of robustness. Heavy reliance on synthetic human motion data (Section 4.1) raises concerns about real-world generalization. The trajectory encoder builds on existing factorized attention [8], and the paper doesn’t fully clarify its novelty over prior video transformer work. The joint task, while innovative, feels like an incremental step given related multi-frame efforts.

---

> ### Author Rebuttal · Authors · 2025-07-31
>
> Dear reviewer p1Bw,
>
> Thank you for your valuable feedback and very detailed comments. We appreciate your remarks on the strengths of our paper, including the task design of joint point mapping and trajectory prediction, the architectural designs, and the strong potential for 3D vision applications. We will address your concerns and questions in the response below.
>
> Note: [x] indicates the reference from the manuscript.
>
> ---
>
> **Why does Track3R underperform test-time optimization methods in point mapping? What specific limitations (e.g., model capacity, training data) cause this?**
>
> We first emphasize that post-optimization methods require heavy computational costs, e.g., all of them considered in our paper demonstrate FPS less than 1, while ours demonstrate 23.81. While such a post-optimization algorithm may improve absolute relative errors (by paying a huge cost), which indicates the (static) point mapping quality, it inevitably biases the final outputs to ignore the motion, which sacrifices the motion estimation performance, such as EPE3D. In more detail, the post-optimization baselines  [2, 3, 16, 19] focus only on static depth maps derived from 3D point maps and minimize the warping errors between different frames. This may smooth out outliers at a specific frame (such as noisy points around the object edges in Figure 1), but can also lose temporal correspondence between the points.
>
> Employing more training data can be a useful solution to further boost the performance of our method. As also suggested by the reviewers YJq8 and bXsD, we have conducted an experiment which employs non-human stereo videos taken with VR180 cameras, along with the auto-labeling pipeline that generates 3D trajectory sequence [17]. Specifically, we train our model with extra 30k clips obtained from raw VR180 videos downloaded from YouTube, and test the performance in the same benchmark configuration as Table 1. The experiment indicates that our training with the additional data could provide overall improvements in the performance, e.g., relative 10.2% enhancement in EPE3D (0.431 $\to$ 0.387 in iPhone) as follows. We note that we are in progress to further scale up the training dataset using more videos from YouTube, and will include the results in the final manuscript.
>
> \begin{array}{c | c  c |  c  c |  c  c }
> \hline
> & \rlap{~~~\text{iPhone}} & \phantom{a} & \rlap{~~~~\text{Sintel}} & \phantom{a} & \rlap{\text{Point Odyssey}} & \phantom{a}  \newline
> \text{Method} & \phantom{a}  \text{EPE}\_\text{3D} \phantom{a} & \phantom{a} \text{Abs-rel} \phantom{a} & \phantom{a} \text{EPE}\_\text{3D} \phantom{a} & \phantom{a} \text{Abs-rel} \phantom{a} & \phantom{a} \text{EPE}\_\text{3D} \phantom{a} & \phantom{a} \text{Abs-rel} \phantom{a}  \newline
> \hline
> \begin{subarray}{c}\text{Fast3R}\end{subarray} & 0.692 & 0.419 & 0.667 & 0.517 & 0.771 & 0.214   \newline
> \begin{subarray}{c}\text{Spann3R}\end{subarray}& 0.658 & 0.431 & 0.523 & 0.622 & 0.591 & 0.231   \newline
> \begin{subarray}{c}\text{CUT3R}\end{subarray} &  0.701 & 0.408 & 0.681 & 0.428 & 0.609 & 0.177   \newline
> \begin{subarray}{c}\text{Track3R}\end{subarray} & 0.431 & 0.344 & 0.312 & 0.374 & 0.399 & 0.165   \newline
> \hline
> \begin{subarray}{c}\textbf{Track3R} \newline
> \textbf{(with VR180)}\end{subarray} & \textbf{0.387} & \textbf{0.323} & \textbf{0.292} & \textbf{0.358} & \textbf{0.371} & \textbf{0.159}  \newline
> \hline
> \end{array}
>
> ---
>
> **Synthetic human motion data may not reflect diverse real-world scenes. Can you provide evidence of performance on varied, non-synthetic data?**
>
> We note that the iPhone dataset benchmark in our experiments is already of the real-world scene, equipped with the sensory 3D point annotations for testing. While our method trained on the synthetic human motion data already demonstrates reasonable performance on this benchmark, we also would like to emphasize that our experiments with the YouTube VR180 data can further boost the performance in this benchmark covering real-world scenes (see the table above).

---

> > ### Comment · Reviewer_p1Bw · 2025-08-04
> >
> > Thank you for the rebuttal. The additional experiments with non-synthetic VR180 data address my concerns about generalization beyond synthetic human motion, and the clarification on trade-offs with test-time optimization methods (e.g., FPS and motion preservation) makes sense, though I still encourage broader real-world benchmarks in the final paper and list the limitations accordingly.
> > Overall, the rebuttal strengthens the paper's case. I am modifying my score accordingly

---

### Official Review · Reviewer_bXsD · 2025-06-27

**Clarity:** 2
**Significance:** 3
**Originality:** 2
**Rating:** 4
**Confidence:** 4

**Summary:**

This work addresses the under constrained problem of 3D reconstruction using 2D monocular videos. Prior works have successfully provided shapes of objects and camera parameters. However, they struggle in cases with object deformation, and their architectures are compatible only for processing pairs of frames, making it difficult to capture complex motion. This work proposes Track3R, which jointly predicts point map (camera poses) and motion trajectories across multiple frames (motion trajectories). The experimental results show 25.8% improvement for motion estimation and 15.7% for point mapping.

**Questions:**

In addition to the "Strengths And Weaknesses" section, does tuning with human motion bias the model to improved performance for Human tracking. Do you observe it affecting the performance of other objects?

**Ethical Concerns:**

["NO or VERY MINOR ethics concerns only"]

**Final Justification:**

The updated computation parameters, the clarification on the novel integration of factorized attention, and the addition of experiments on non-human subjects have all effectively addressed my previous queries. I am maintaining my positive score for the paper.

**Limitations:**

Yes,

I think additional discussion/ablations on non-human object tracking would help get a better understanding of the approach.

**Paper Formatting Concerns:**

No major formatting issues.

**Quality:**

3

**Strengths And Weaknesses:**

**Strengths**

1. The model's ability to process multiple frames is a valuable contribution to the challenging problem of 3D understanding.

2. The reported performance gains are significant, with a 25.8% improvement in 3D motion estimation and a 15.7% provements in point mapping accuracy.

**Weaknesses**

1. Computation comparison are needed (#parameters, #GPU hours, training and inference times).

2. ⁠The contributions are not clear (atleast not clearly stated).
     a.  Adapting to multiple frames comes from [8].
     b. The main novelty is the two path simiese for disentangling motion and camera

3. Sec 3.1 and Sec3.2 are a bit difficult to understand because of the notations.  X^{i|j} which represents the 3D coordinate of frame I^i and does not involve j confuses the part. I think it gets cleared later on to learn motion but better notations would help.

4. Making the captions self-sufficient for Fig1. will improve the quality of the paper.







[8] Gedas Bertasius, Heng Wang, and Lorenzo Torresani. Is space-time attention all you need for videounderstanding? In ICML, page 4, 2021.

---

> ### Author Rebuttal · Authors · 2025-07-31
>
> Dear reviewer bXsD,
>
> Thank you for your valuable feedback and very detailed comments. We appreciate your remarks on the strengths of our paper, including the multi-frame processing designs, and the empirical results. We will address your concerns and questions in the response below.
>
> Note: [x] indicates the reference from the manuscript.
>
> ---
>
> **Computation comparisons.**
>
> We note that the comparison for inference speed (frame per second; FPS) is available in Table 1, for example, ours can demonstrate 23.81 while post-optimization baselines demonstrate FPS less than 1.
>
> Other computation factors, such as #parameters, #GPU hours (training time), we provide the comparison to the best of our knowledge, referring to the papers and the open-source repositories of the baselines [4, 20, 22], as follows.
>
> \begin{array}{c | c  c  c c}
> \hline
> \text{Method} & \text{GPU Type} & \text{\\# parameters} & \text{GPU hours} & \text{Training}  \newline
> \hline
> \begin{subarray}{c}\text{Fast3R}\end{subarray} & \text{A100} & \text{648M} & - &  \text{from scratch}  \newline
> \begin{subarray}{c}\text{Spann3R}\end{subarray} & \text{V100} & \text{803M} & 1920 & \text{fine-tuning from DuST3R}   \newline
> \begin{subarray}{c}\text{CUT3R}\end{subarray} & \text{A100} & \text{665M} & 18831 & \text{from scratch}   \newline
> \begin{subarray}{c}\textbf{Track3R (ours)}\end{subarray} & \text{A100} & \text{688M} & 288 & \text{fine-tuning from MonsT3R}   \newline
> \hline
> \end{array}
>
> ---
>
> **Remarks on the architectural novelty and key contributions.**
>
> We would like to emphasize that the factorized attention [8] is a part of our trajectory encoder module to enable information sharing over multiple frames. However, the merit of our architecture design, as recognized by the reviewers XKzt and p1Bw, is our specific way of integrating the module to the base architecture and inflating the pairwise model towards the multi-frame processing, which effectively retains strong prior in a pre-trained model.
>
> In addition to the architectural novelty, we also would like to note that we tackle the scarcity of 4D training data for training. As pointed out as the strength of our paper by the reviewer YJq8, our method of leveraging ground-truth motion data from synthesized human action videos provides a significant effect on the point mapping and trajectory prediction performance, e.g., Table 3.
>
> ---
>
> **Terminology and editorial suggestions.**
>
> Thank you for pointing out the ambiguity of $X^{i|j}$, where its semantics may vary depending on whether it is used under the context of pairwise baselines, or the context of our method which incorporates motion over the frames. We will revise the presentation, and make a further effort to clarify the terminology. We will also elaborate more on the captions for Figure 1, e.g., include the definition of the trajectory prediction ($X$) and the point mapping ($Y$), and the difference between our method versus the pairwise models.
>
> ---
>
> **Experiment with non-human object data.**
>
> As also suggested by the reviewer YJq8, we have conducted an experiment which employs non-human stereo videos taken with VR180 cameras, along with the auto-labeling pipeline that generates 3D trajectory sequence [17]. Specifically, we train our model with extra 30k clips obtained from raw VR180 videos downloaded from YouTube, and test the performance in the same benchmark configuration as Table 1. The experiment indicates that our training with the additional data could provide overall improvements in the performance, e.g., relative 10.2% enhancement in EPE3D (0.431 $\to$ 0.387 in iPhone) as follows.
>
> \begin{array}{c | c  c |  c  c |  c  c }
> \hline
> & \rlap{~~~\text{iPhone}} & \phantom{a} & \rlap{~~~~\text{Sintel}} & \phantom{a} & \rlap{\text{Point Odyssey}} & \phantom{a}  \newline
> \text{Method} & \phantom{a}  \text{EPE}\_\text{3D} \phantom{a} & \phantom{a} \text{Abs-rel} \phantom{a} & \phantom{a} \text{EPE}\_\text{3D} \phantom{a} & \phantom{a} \text{Abs-rel} \phantom{a} & \phantom{a} \text{EPE}\_\text{3D} \phantom{a} & \phantom{a} \text{Abs-rel} \phantom{a}  \newline
> \hline
> \begin{subarray}{c}\text{Fast3R}\end{subarray} & 0.692 & 0.419 & 0.667 & 0.517 & 0.771 & 0.214   \newline
> \begin{subarray}{c}\text{Spann3R}\end{subarray}& 0.658 & 0.431 & 0.523 & 0.622 & 0.591 & 0.231   \newline
> \begin{subarray}{c}\text{CUT3R}\end{subarray} &  0.701 & 0.408 & 0.681 & 0.428 & 0.609 & 0.177   \newline
> \begin{subarray}{c}\text{Track3R}\end{subarray} & 0.431 & 0.344 & 0.312 & 0.374 & 0.399 & 0.165   \newline
> \hline
> \begin{subarray}{c}\textbf{Track3R} \newline
> \textbf{(with VR180)}\end{subarray} & \textbf{0.387} & \textbf{0.323} & \textbf{0.292} & \textbf{0.358} & \textbf{0.371} & \textbf{0.159}  \newline
> \hline
> \end{array}

---

> > ### Comment · Reviewer_bXsD · 2025-08-05
> >
> > I appreciate the response from the authors. The updated **computation parameters**, **the clarification on the novel integration of factorized attention**, and the **addition of experiments on non-human subjects** have all effectively addressed my previous queries. I am maintaining my positive score for the paper. Thanks!

---

### Official Review · Reviewer_XKzt · 2025-07-03

**Clarity:** 4
**Significance:** 3
**Originality:** 3
**Rating:** 5
**Confidence:** 4

**Summary:**

This paper introduces Track3R, a novel framework for predicting point maps and trajectories from monocular videos. It makes two primary contributions: First, it proposes a new formulation that jointly predicts a 3D point map (representing object shape relative to a moving camera) and a 3D motion trajectory (representing object dynamics in a fixed reference frame). Second, it introduces a novel "Trajectory Encoder" module, based on factorized temporal attention, which allows a Siamese-style architecture to efficiently process multiple frames in a single forward pass while retaining valuable priors from pre-trained models. The authors demonstrate through extensive experiments that Track3R significantly improves performance on joint 3D motion estimation and point mapping tasks in dynamic scenes, outperforming feed-forward baselines and achieving competitive results with much slower test-time optimization methods.

**Questions:**

I am willing to raise my score if the points mentioned in the 'Weaknesses' section are addressed satisfactorily.

**Ethical Concerns:**

["NO or VERY MINOR ethics concerns only"]

**Final Justification:**

This paper presents a solid and technically sound method for joint 3D motion estimation and point mapping tasks, and it achieves better performance than other methods. In the rebuttal, the authors provided more comprehensive experimental comparisons with other methods and discussed more related works. This has addressed my concerns, and I therefore recommend "accept" for this paper.

**Limitations:**

Yes

**Quality:**

3

**Strengths And Weaknesses:**

# Strength
1. Clear Problem Formulation: The core idea of disentangling camera motion from object motion by jointly predicting point maps and 3D trajectories is clear and well-motivated. This is achieved by repurposing one of the decoder's outputs to represent the 3D coordinates of a point at different time indices but from a fixed camera view, effectively defining its 3D motion trajectory.


2. Elegant Architectural Design: The proposed Trajectory Encoder is a clever architectural modification. The use of factorized temporal attention to process multiple frames is efficient. Crucially, the integration via a layerscale initialized to a small value is an insightful technique to "inflate" a pre-trained pairwise model for a multi-frame task without suffering from catastrophic forgetting.


3. Impressive Empirical Results: The experimental results are strong and convincing. Track3R shows marked improvements over state-of-the-art feed-forward methods in both motion estimation (EPE_3D) and point mapping (Abs_rel). The fact that it achieves this with a high inference speed (23.81 FPS) and is competitive with methods requiring costly test-time optimization is a significant practical advantage.


4. High Clarity and Readability: The paper is well-written, logically structured, and easy to follow. The motivation is clearly laid out, the method is described in sufficient detail, and the notation is clear, making the method easy to understand.



# Weakness

1. Comparison with 3D Tracking Methods: While the proposed method shows impressive performance on jointly predicting point maps and tracking trajectories, a comparison with existing 3D tracking methods (e.g., DELTA [1], TAP3D [2], SpatialTracker [3]) on standard tracking metrics would greatly help readers understand how Track3R compares to them.

2. Insufficient Related Work: There are other works that also focus on jointly predicting point maps and tracking, such as GFlow [4], POMATO [5], St4RTrack [6] which should be discussed in the related work section.

3. Suggestions for Presentation:
* The term "shape" could be more appropriately replaced with "geometry" throughout the paper for better technical precision.
* The aesthetics of Figure 2 could be improved (e.g., by avoiding a high-contrast color palette).
* In Line #20, the statement "The recent advance of 3D prior models" should be followed by citations to the relevant works.


[1] DELTA: Dense Efficient Long-range 3D Tracking for Any video
[2] TAPIP3D: Tracking Any Point in Persistent 3D Geometry
[3] SpatialTracker: Tracking Any 2D Pixels in 3D Space
[4] GFlow: Recovering 4D World from Monocular Video
[5] POMATO: Marrying Pointmap Matching with Temporal Motion for Dynamic 3D Reconstruction
[6] St4RTrack: Simultaneous 4D Reconstruction and Tracking in the World

---

> ### Author Rebuttal · Authors · 2025-07-31
>
> Dear reviewer XKzt,
>
> Thank you for your valuable feedback and very detailed comments. We appreciate your remarks on the strengths of our paper, including the clear problem formulation and presentation, the architectural designs, and the empirical results. We will address your concerns and questions in the response below.
>
> Note: [x] indicates the reference from the manuscript; [ref-x] indicates extra references for the rebuttal.
>
> ---
>
> **Comparison with 3D Tracking Methods (TAPIP3D, SpatialTracker, DELTA)**
>
> For the comparison, we allow 3D tracking models [ref-3, ref-4, ref-5] to access ground truth camera poses and simulate coordinate transforms between video frames (we note that the 3D tracking does not support point mapping functions). Since our focus is modeling 3D trajectory of pixel points in a free world space (without predicting occlusions with respect to a specific camera pose and frame pixels), we choose to experiment with the configuration of Table 1.
>
> Overall, the 3D tracking methods demonstrate relatively better motion estimation performance (EBE3D), similar to our method, than one-pass point mapping baselines [4, 20, 22]. However, the 3D tracking methods cannot match the point mapping performance (Abs-rel), even if the ground truth camera poses are provided. We argue these results are anticipated since the 3D tracking models are directly supervised for motion estimation, while being unable to learn strong 3D geometry prior from static 3D point mapping data.
>
> \begin{array}{c | c | c  c |  c  c |  c  c }
> \hline
> & &  \rlap{~~~\text{iPhone}} & \phantom{a} & \rlap{~~~~\text{Sintel}} & \phantom{a} & \rlap{\text{Point Odyssey}} & \phantom{a}  \newline
> \text{Model Type} & \text{Method} & \phantom{a}  \text{EPE}\_\text{3D} \phantom{a} & \phantom{a} \text{Abs-rel} \phantom{a} & \phantom{a} \text{EPE}\_\text{3D} \phantom{a} & \phantom{a} \text{Abs-rel} \phantom{a} & \phantom{a} \text{EPE}\_\text{3D} \phantom{a} & \phantom{a} \text{Abs-rel} \phantom{a}  \newline
> \hline
> & \begin{subarray}{c}\text{TAPIP3D}\end{subarray} & 0.499 & 0.717 & 0.331 & 0.626 & 0.441 & 0.636   \newline
> \text{3D Tracking} & \begin{subarray}{c}\text{Spatial Tracker}\end{subarray}& 0.443 & 0.699 & 0.339 & 0.582 & 0.422 & 0.791   \newline
>  & \begin{subarray}{c}\text{DELTA}\end{subarray} &  \underline{0.434} & 0.501 & \underline{0.320} & 0.611 & \textbf{0.397} & 0.244   \newline
> \hline
> & \begin{subarray}{c}\text{Fast3R}\end{subarray} & 0.692 & 0.419 & 0.667 & 0.517 & 0.771 & 0.214   \newline
> \text{Point} & \begin{subarray}{c}\text{Spann3R}\end{subarray}& 0.658 & 0.431 & 0.523 & 0.622 & 0.591 & 0.231   \newline
> \text{Mapping} & \begin{subarray}{c}\text{CUT3R}\end{subarray} &  0.701 & \underline{0.408} & 0.681 & \underline{0.428} & 0.609 & \underline{0.177}   \newline
> & \begin{subarray}{c}\textbf{Track3R (ours)}\end{subarray} & \textbf{0.431} & \textbf{0.344} & \textbf{0.312} & \textbf{0.374} & \underline{0.399} & \textbf{0.165}   \newline
> \hline
> \end{array}
>
> [ref-3] Zhang et al., "TAPIP3D: Tracking Any Point in Persistent 3D Geometry" (2025)
>
> [ref-4] Xiao et al., "SpatialTracker: Tracking Any 2D Pixels in 3D Space", CVPR 2024
>
> [ref-5] Ngo et al., "DELTA: Dense Efficient Long-range 3D Tracking for Any video", ICLR 2025
>
> ---
>
>
> **Concurrent works and editorial suggestions.**
>
> Thank you for pointing out GFlow [ref-6], POMATO [ref-7], St4RTrack [ref-8], which are concurrent works that aim similar tasks as our method, which we will include in our updated manuscript. While these models partially share similar design as ours, we will highlight the key differences and discuss in detail. For example, the Gaussian Splatting in GFlow  [ref-6] vs. one-pass prediction in ours, the separate temporal head for motion estimation in POMATO [ref-7] vs. the modification of the semantics of point mapping $X$ in ours, and the pairwise processing architecture in St4RTrack [ref-8] vs. the multi-frame processing architecture in ours.
>
> We will also clarify and revise the suggested terminology and improve the color palette of the figures.
>
> [ref-6] Wang et al., "GFlow: Recovering 4D World from Monocular Video" AAAI 2025
>
> [ref-7] Zhang et al., "POMATO: Marrying Pointmap Matching with Temporal Motion for Dynamic 3D Reconstruction" (2025)
>
> [ref-8] Feng et al., "St4RTrack: Simultaneous 4D Reconstruction and Tracking in the World", ICCV 2025

---

> > ### Comment · Reviewer_XKzt · 2025-08-05
> >
> > Thank you for the detailed rebuttal. My concerns are now well addressed, and I have raised the score accordingly.

---

### Official Review · Reviewer_YJq8 · 2025-07-04

**Clarity:** 2
**Significance:** 3
**Originality:** 3
**Rating:** 4
**Confidence:** 4

**Summary:**

To address the limitations of previous work focused on static scenes and pairwise processing, this paper introduces a novel framework that jointly predicts both the point map and point trajectories. The authors enhance the model architecture to handle multi-frame prediction, improving efficiency over post-optimization-based methods. To obtain ground-truth point trajectories, they leverage a ground-truth human motion dataset and render videos as input. Experimental results demonstrate the effectiveness of the proposed method compared to prior approaches.

**Questions:**

As noted in the weaknesses, there is a concern that the current point trajectory requirement may be too strict and could introduce ambiguity. It would be valuable to see how the performance changes if post-optimization is also incorporated, along with some discussion comparing this to the VGGT paper.

**Ethical Concerns:**

["NO or VERY MINOR ethics concerns only"]

**Final Justification:**

Thank you to the authors for the responses. The answers addressed my questions. The proposed method can further motivate the community to explore one-pass point mapping for dynamic scenes. I will keep my original score.

**Limitations:**

yes

**Quality:**

3

**Strengths And Weaknesses:**

Strengths
1. The paper reframes the 4D understanding problem as a joint task of point map and trajectory motion prediction. By incorporating a design for multi-frame processing, the proposed framework achieves improved 4D video understanding, enhancing both motion estimation and mapping accuracy and efficiency.
2. The approach of leveraging ground-truth motion data and synthetic videos is a reasonable way to address the challenge of obtaining ground-truth point trajectories, and its effectiveness is demonstrated through the experiments.
3. The paper further demonstrates the potential of one-pass multi-frame prediction to replace the need for post-optimization, although there is still room for improvement.

Weaknesses
1. It is inherently difficult to obtain ground-truth point trajectories for arbitrary videos, since objects or parts that appear in later frames may not be visible in the initial frames, leading to ambiguity. Why not incorporate a visibility mask to indicate whether points are present in each frame, similar to the design in CoTracker? By relaxing this constraint, it may be possible to leverage more data for the joint task.
2. Most post-optimization-based methods achieve better absolute relative error. However, it’s unclear what the core challenge is in diverging from the pseudo ground truth—does it primarily stem from the strict requirement of complete point trajectories? Would the results of Track3R also improve if combined with post-optimization, and what would the performance look like after such refinement?
3. The paper does not discuss topics related to VGGT. Although VGGT was originally designed for static scenes, it would be helpful to understand the advantages and disadvantages of the proposed method in handling multi-frame prediction compared to VGGT.

---

> ### Author Rebuttal · Authors · 2025-07-31
>
> Dear reviewer YJq8,
>
> Thank you for your valuable feedback and very detailed comments. We appreciate your remarks on the strengths of our paper, including the task design of joint point mapping and trajectory prediction, one-pass multi-frame prediction framework, and employing motion data and synthetic videos for training. We will address your concerns and questions in the response below.
>
> Note: [x] indicates the reference from the manuscript; [ref-x] indicates extra references for the rebuttal.
>
> ---
>
> **Why not incorporate a visibility mask to indicate whether points are present in each frame? By relaxing this constraint, it may be possible to leverage more data for the joint task.**
>
>
> To begin, we would like to clarify that our method can incorporate visibility masks in training data, and our point mapping function (e.g., mapping objects in later frames to the camera view of an initial frame) already utilizes them for training, which denotes ambiguous or degenerate regions induced by the change of camera views among different frames. For instance, these visibility masks are employed when calculating the confidence-aware and scale-invariant regression loss [2, 3].
>
> It is also worth noting that the visibility mask can help leverage more training data. For your interest, we have conducted an experiment which employs unlabeled stereo videos taken with VR180 cameras, along with the auto-labeling pipeline that generates 3D trajectory sequence and visibility masks [17]. Specifically, we train our model with extra 30k clips obtained from raw VR180 videos downloaded from YouTube, and test the performance in the same benchmark configuration as Table 1. The experiment indicates that our training with the additional data could provide overall improvements in the performance, e.g., relative 10.2% enhancement in EPE3D (0.431 $\to$ 0.387 in iPhone) as follows.
>
> \begin{array}{c | c  c |  c  c |  c  c }
> \hline
> & \rlap{~~~\text{iPhone}} & \phantom{a} & \rlap{~~~~\text{Sintel}} & \phantom{a} & \rlap{\text{Point Odyssey}} & \phantom{a}  \newline
> \text{Method} & \phantom{a}  \text{EPE}\_\text{3D} \phantom{a} & \phantom{a} \text{Abs-rel} \phantom{a} & \phantom{a} \text{EPE}\_\text{3D} \phantom{a} & \phantom{a} \text{Abs-rel} \phantom{a} & \phantom{a} \text{EPE}\_\text{3D} \phantom{a} & \phantom{a} \text{Abs-rel} \phantom{a}  \newline
> \hline
> \begin{subarray}{c}\text{Fast3R}\end{subarray} & 0.692 & 0.419 & 0.667 & 0.517 & 0.771 & 0.214   \newline
> \begin{subarray}{c}\text{Spann3R}\end{subarray}& 0.658 & 0.431 & 0.523 & 0.622 & 0.591 & 0.231   \newline
> \begin{subarray}{c}\text{CUT3R}\end{subarray} &  0.701 & 0.408 & 0.681 & 0.428 & 0.609 & 0.177   \newline
> \begin{subarray}{c}\text{Track3R}\end{subarray} & 0.431 & 0.344 & 0.312 & 0.374 & 0.399 & 0.165   \newline
> \hline
> \begin{subarray}{c}\textbf{Track3R} \newline
> \textbf{(with VR180)}\end{subarray} & \textbf{0.387} & \textbf{0.323} & \textbf{0.292} & \textbf{0.358} & \textbf{0.371} & \textbf{0.159}  \newline
> \hline
> \end{array}
>
> ---
>
> **Would the results of Track3R also improve if combined with post-optimization, and what would the performance look like after such refinement?**
>
> While the post-optimization algorithm may improve absolute relative errors, which indicates the (static) point mapping quality, it inevitably biases the final outputs to ignore the motion, which sacrifices the motion estimation performance, such as EPE3D. It also requires heavy computational costs (e.g., all post-optimization baselines demonstrate FPS less than 1, while ours demonstrate 23.81).
>
> In more detail, the post-optimization algorithm in the baselines [2, 3, 16, 19] focus only on static depth maps derived from 3D point maps and minimize the warping errors between different frames. This may smooth out outliers at a specific frame (such as noisy points around the object edges in Figure 1), but can also lose temporal correspondence between the points.
>
> We would like to emphasize that our focus is obtaining motion and geometry prior for dynamic scenes, orthogonal to post-optimization algorithms, hence we mainly compare our model to the one-step prediction baselines. For example, we compare the camera pose estimation results (Table 2), which are obtained by applying the iterative least-squares optimization to the outputs of the one-step predictions, following the experimental configuration of CUT3R [4].
>
> ---
>
> **Comparison to VGGT.**
>
> Both ours and VGGT [ref-1] modify the Siamese architecture from prior art to obtain one-pass point mapping models. However, in addition to the key difference in their task configuration, i.e., dynamic scene (ours) vs. static scene (VGGT), another notable difference between the two models is that VGGT directly modifies the cross-frame attentions within the base Siamese architecture for point mapping, while ours introduces the trajectory encoder and avoid modifying the attention layer to prevent catastrophic forgetting in the pre-trained model.
>
> Instead, VGGT chooses to encode image features using a foundation vision model (e.g., DINO [ref-2]) to provide a strong prior for 3D tasks to their model, and focuses on fine-tuning the model for the point mapping task. Since DINO can act as a strong and robust representation for static images, it significantly contributes to VGGT demonstrating state-of-the-art performance on static scene configurations. We believe their approach, employing image foundation models for 3D tasks is a reasonable direction for the point mapping task, and extending to dynamic scenes, such as employing video foundation models, can be an interesting future work.
>
> [ref-1] Wang et al., "VGGT: Visual Geometry Grounded Transformer", CVPR 2025
>
> [ref-2] Oquab et al., "DINOv2: Learning Robust Visual Features without Supervision", Transactions on Machine Learning Research (2024)

---

> > ### Comment · Reviewer_YJq8 · 2025-08-05
> >
> > Thanks to the authors for the responses. The answers addressed my questions. The proposed method can further motivate the community to explore one-pass point mapping for dynamic scenes. I will keep my original score.

---

### Decision · Program_Chairs · 2025-09-17

**Decision:**

Accept (poster)

**Comment:**

Track3R delivers a technically sound and efficient approach to spatiotemporal 3D understanding in dynamic scenes, with clear gains over feed-forward baselines and a clean path to further scaling. For the camera-ready, I encourage the authors to: (i) tighten claims around point-mapping vs optimization, (ii) further clarify notation/figures and adopt “geometry” consistently, and (iii) expand related-work positioning/comparison (VGGT, GFlow, POMATO, TAPIP3D, St4RTrack) and explicitly list limitations and compute trade-offs.  (IV) include more real-world benchmark comparison as Reviewer p1Bw mentioned and also considering the TAPVid-3D benchmark.